# Data Distillation: A Survey

**Noveen Sachdeva**  *nosachde@ucsd.edu*
*Computer Science & Engineering*
*University of California, San Diego*

**Julian McAuley**  *jmcauley@ucsd.edu*
*Computer Science & Engineering*
*University of California, San Diego*

**Reviewed on OpenReview:** *https://openreview.net/forum?id=lmXMXP74TO*

## Abstract

The popularity of deep learning has led to the curation of a vast number of massive and multifarious datasets. Despite having close-to-human performance on individual tasks, training parameter-hungry models on large datasets poses multi-faceted problems such as (a) high model-training time; (b) slow research iteration; and (c) poor eco-sustainability. As an alternative, *data distillation* approaches aim to synthesize terse data summaries, which can serve as effective drop-in replacements of the original dataset for scenarios like model training, inference, architecture search, *etc.* In this survey, we present a formal framework for data distillation, along with providing a detailed taxonomy of existing approaches. Additionally, we cover data distillation approaches for different data modalities, namely images, graphs, and user-item interactions (recommender systems), while also identifying current challenges and future research directions.

## 1 Introduction

**(Loose) Definition 1. (Data distillation)** *Approaches that aim to synthesize tiny and high-fidelity data summaries which distill the most important knowledge from a given target dataset. Such distilled summaries are optimized to serve as effective drop-in replacements of the original dataset for efficient and accurate data-usage applications like model training, inference, architecture search, etc.*

The recent "scale-is-everything" viewpoint (Ghorbani et al., 2021; Hoffmann et al., 2022; Kaplan et al., 2020), argues that training bigger models (*i.e.*, consisting of a higher number of parameters) on bigger datasets, and using larger computational resources is the sole key for advancing the frontier of artificial intelligence. Such studies observe and hypothesize the generalizability of neural networks as a power-law *w.r.t.* the aforementioned factors, albeit with small exponents. On the other hand, a reasonable argument is that a principled and well-reasoned solution will be more amenable to various scaling-laws, thereby leading to faster progress. Data distillation (Definition 1) is clearly a task rooted in the latter school of thought by introducing the *fidelity of data* as in important covariate in such neural scaling-laws. Sorscher et al. (2022) demonstrate this viewpoint analytically by using simple heuristics to prune away data with low measures of signal for model training. Clearly, the scale viewpoint still holds, in that if we keep increasing the amount of data (albeit now compressed and of higher quality), we will observe an improvement in both upstream and downstream generalization, but at a faster rate.

**Motivation.** A terse, high-quality data summary has use cases from a variety of standpoints. First and foremost, it leads to a faster model-training procedure. In turn, faster model training equates to (1) compute-cost saving and expedited research iterations, *i.e.*, the investigative procedure of manually experimenting different ideas; and (2) improved eco-sustainability, *i.e.*, lowering the amount of compute time directly leads to a lower carbon footprint from running power-hungry accelerated hardware (Gupta et al., 2022). Additionally,

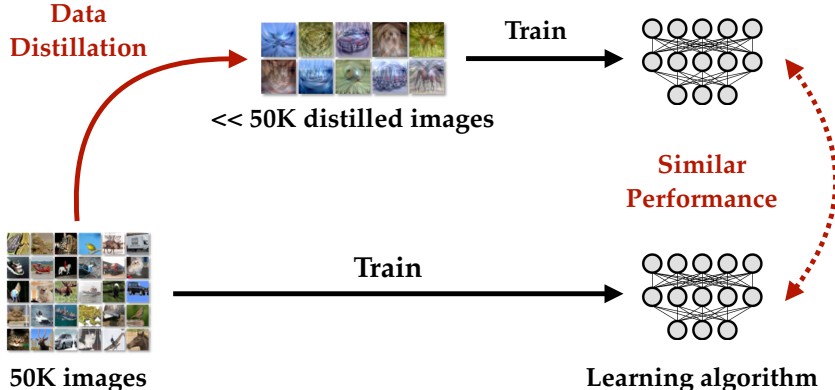

[HQ Image Link] Figure 1: The premise of data distillation demonstrated using an image dataset.

a small data summary democratizes the entire pipeline, as more people can train state-of-the-art algorithms on reasonably accessible hardware using the data summary. Finally, a high-quality data summary indirectly also accelerates orthogonal procedures like neural architecture search (Liu et al., 2019), approximate nearest neighbour search (Arya et al., 1998), knowledge distillation (Hinton et al., 2015), *etc.*, where the procedure needs to iterate over the entire dataset multiple times.

**Comparison with data pruning.** Another reasonable avenue for summarizing large datasets is *pruning* away *low-quality* data which presumably does not carry large amount of *signal* to be captured during model-training. The primary challenge for such data pruning approaches (*a.k.a.* coreset construction) lies in tagging the *hardness* of each data-point which can be used for subsequent pruning (typically in a greedy fashion). Prominent data-pruning approaches propose heuristics for the same, relying on concepts such as shapley values (Ghorbani & Zou, 2019), confidence scores (Coleman et al., 2020), error-contribution (Toneva et al., 2019), feature-space geometry (Abbas et al., 2023; Sorscher et al., 2022; Welling, 2009), *etc.* Another line of work builds on the advances in submodular optimization (see Bilmes (2022) for a review) to approximately solve the NP-Hard combinatorial optimization of selecting the subset that maximizes a set-level *goodness* function, when such goodness functions are provably submodular (Killamsetty et al., 2021; Mirzasoleiman et al., 2020; S et al., 2021). Notably, such data pruning methodologies inherently share the same goal as data distillation but are severely restricted due to only retaining data already in the target dataset, thereby leading to finite expressivity and hence, generally, lower sample-fidelity (see Ayed & Hayou (2023) for a deeper theoretical outlook on the fundamental limitations of data pruning). Further, recent empirical studies of data pruning methodologies (Guo et al., 2022) demonstrate that the efficacy of such data pruning heuristics rarely and irregularly transfers to practical scenarios, with random downsampling being a hard baseline.

**Comparison with knowledge distillation & transfer learning.** Despite inherently distilling some notion of *knowledge*, we would like to highlight both *knowledge distillation* and *transfer learning* are orthogonal procedures to data distillation, and can potentially work in conjunction to perform both tasks more efficiently. More specifically, knowledge distillation (Hinton et al., 2015) entails distilling the knowledge from a trained teacher network into a smaller student network. On the other hand, transfer learning (Pratt, 1992) focuses on transferring knowledge across similar tasks, *e.g.*, from image classification to image segmentation. Orthogonally, data distillation aims to distill the knowledge from a given dataset into a terse data summary. Such data summaries can be used *in conjunction* with knowledge distillation or transfer learning procedures for both (1) faster learning of the teacher models; and (2) faster knowledge transfer to the student models. The same holds true for model compression techniques (LeCun et al., 1989), where similar to knowledge distillation, the goal is to reduce model storage size rather than reducing the training time or increasing the sample-fidelity.

*In this survey*, we intend to provide a succinct overview of various data distillation frameworks across different data modalities. We start by presenting a formal data distillation framework in Section 2, and present technicalities of various existing techniques. We classify all data distillation techniques into four categories

(see Figure 2 for a taxonomy) and provide a detailed empirical comparison of image distillation techniques in Table 1. Subsequently, in Section 3, we discuss existing data distillation frameworks for synthesizing data of different modalities, as well as outlining the associated challenges. In Section 4, we discuss alternative applications of synthesizing a high-fidelity data summary rather than simply accelerating model training along with pointers to existing work. Finally, in Section 5, we conclude by presenting common pitfalls in existing data distillation techniques, along with proposing interesting directions for future work.

## 2 The Data Distillation Framework

Before going into the specifics of data distillation, we start by outlining useful notation. Let $\mathcal{D} \triangleq \{(x_i, y_i)\}_{i=1}^{|\mathcal{D}|}$ be a given dataset which needs to be distilled, where $x_i \in \mathcal{X}$ are the set of input features, and $y_i \in \mathcal{Y}$ is the desired label for $x_i$. For classification tasks, let $\mathcal{C}$ be the set of unique classes in $\mathcal{Y}$, and $\mathcal{D}^c \triangleq \{(x_i, y_i) \mid y_i = c\}_{i=1}^{|\mathcal{D}|}$ be the subset of $\mathcal{D}$ with class $c$. We also define the matrices $\mathbf{X} \triangleq [x_i]_{i=1}^{|\mathcal{D}|}$ and $\mathbf{Y} \triangleq [y_i]_{i=1}^{|\mathcal{D}|}$ for convenience. Given a data budget $n \in \mathbb{Z}^+$, data distillation techniques aim to synthesize a high-fidelity data summary $\mathcal{D}_{\mathsf{syn}} \triangleq \{(\tilde{x}_i, \tilde{y}_i)\}_{i=1}^{n}$ such that $n \ll |\mathcal{D}|$. We define $\mathcal{D}_{\mathsf{syn}}^c$, $\mathbf{X}_{\mathsf{syn}}$, and $\mathbf{Y}_{\mathsf{syn}}$ similarly as defined for $\mathcal{D}$. Let $\Phi_\theta : \mathcal{X} \mapsto \mathcal{Y}$ represent a learning algorithm parameterized by $\theta$. We also assume access to a twice-differentiable cost function $l : \mathcal{Y} \times \mathcal{Y} \mapsto \mathbb{R}$, and define $\mathcal{L}_\mathcal{D}(\theta) \triangleq \mathbb{E}_{(x,y)\sim\mathcal{D}}[l(\Phi_\theta(x), y)]$ for convenience. Notation is also summarized in Appendix A. Notably, since $\mathcal{D}$ and $\mathcal{D}_{\mathsf{syn}}$ share the same data domain ($\mathcal{X}$), under reasonable systems' assumptions, training $\Phi$ using gradient descent (GD) on $\mathcal{D}_{\mathsf{syn}}$ will have a $\frac{|\mathcal{D}|}{n} \times$ training-time speedup compared to training $\Phi$ on $\mathcal{D}$.

For the sake of uniformity, we refer to the data synthesized by data distillation techniques as a *data summary* henceforth. Inspired by the definition of coresets (Bachem et al., 2017), we formally define an $\epsilon-$approximate data summary, and the data distillation task as follows:

**Definition 2. ($\epsilon-$approximate data summary)** *Given a learning algorithm $\Phi$, let $\theta^\mathcal{D}$, $\theta^{\mathcal{D}_{\mathsf{syn}}}$ represent the optimal set of parameters for $\Phi$ estimated on $\mathcal{D}$ and $\mathcal{D}_{\mathsf{syn}}$, and $\epsilon \in \mathbb{R}^+$; we define an $\epsilon-$approximate data summary as one which satisfies:*

$$\sup \; \{ \mid l\left(\Phi_{\theta^\mathcal{D}}(x), y\right) - l\left(\Phi_{\theta^{\mathcal{D}_{\mathsf{syn}}}}(x), y\right) \mid \}_{\substack{x\sim\mathcal{X} \\ y\sim\mathcal{Y}}} \; \leq \; \epsilon \tag{1}$$

**Definition 3. (Data distillation)** *Given a learning algorithm $\Phi$, let $\theta^\mathcal{D}$, $\theta^{\mathcal{D}_{\mathsf{syn}}}$ represent the optimal set of parameters for $\Phi$ estimated on $\mathcal{D}$ and $\mathcal{D}_{\mathsf{syn}}$; we define data distillation as optimizing the following:*

$$\underset{\mathcal{D}_{\mathsf{syn}}, n}{\arg\min} \left( \sup \; \{ \mid l\left(\Phi_{\theta^\mathcal{D}}(x), y\right) - l\left(\Phi_{\theta^{\mathcal{D}_{\mathsf{syn}}}}(x), y\right) \mid \}_{\substack{x\sim\mathcal{X} \\ y\sim\mathcal{Y}}} \right) \tag{2}$$

From Definition 3, we highlight three cornerstones of evaluating data distillation methods: (1) Performance: downstream evaluation of models trained on the synthesized data summary *vs.* the full dataset (*e.g.*, accuracy, FID, nDCG, *etc.*); (2) Efficiency: how quickly can models reach full-data performance (or even exceed it), *i.e.*, the scaling of $n$ *vs.* downstream task-performance; and (3) Transferability: how well can data summaries generalize to a diverse pool of learning algorithms, in terms of downstream evaluation.

**No free lunch.** The universal "No Free Lunch" theorem (Wolpert & Macready, 1997) applies to data distillation as well. For example, looking at the transferability of a data summary, it is strongly dependent on the set of encoded inductive biases, *i.e.*, through the choice of the learning algorithm $\Phi$ used while distilling, as well as the objective function $l(\cdot, \cdot)$. Such biases are unavoidable for any data distillation technique, in a sense that learning algorithms closely following the set of encoded inductive biases, will be able to generalize better on the data summary than others.

Keeping these preliminaries in mind, we now present a formal framework for data distillation, encapsulating existing data distillation approaches. Notably, the majority of existing techniques intrinsically solve a bilevel optimization problem, which are tractable surrogates of Equation (2). The inner-loop typically optimizes a representative learning algorithm on the data summary, and using the optimized learning algorithm, the outer-loop optimizes a tractable proxy of Equation (2).

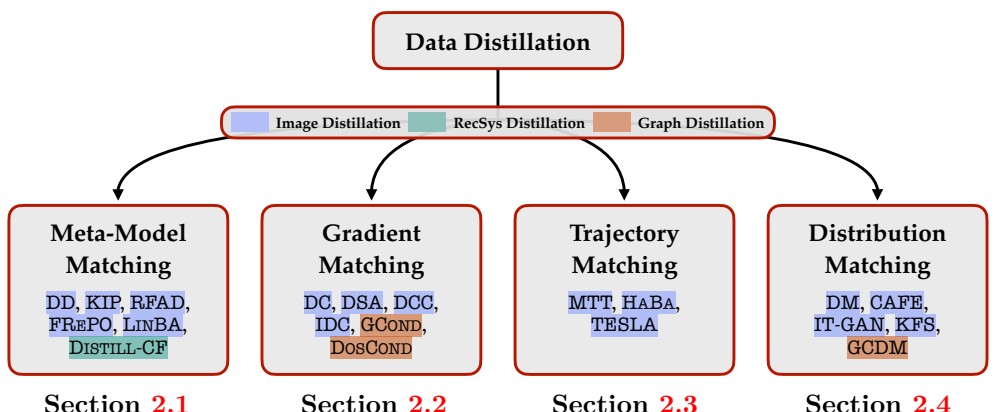

[HQ Image Link] Figure 2: A taxonomy of existing data distillation approaches.

Some common assumptions that existing data distillation techniques follow are: (1) static-length data summary, *i.e.*, $n$ is fixed and is treated as a tunable hyper-parameter; and (2) we have on-demand access to the target dataset $\mathcal{D}$ which is also assumed to be `iid`. Notably, the outer-loop optimization of $\mathcal{D}_{\mathsf{syn}}$ happens simply through GD on the analogously defined $\mathbf{X}_{\mathsf{syn}} \in \mathbb{R}^{n \times \dim(\mathcal{X})}$, which is instantiated as free parameters. Note that the labels, $\mathbf{Y}_{\mathsf{syn}} \in \mathbb{R}^{n \times \dim(\mathcal{Y})}$, can be similarly optimized through GD as well (Bohdal et al., 2020). For the sake of notational clarity, we will interchangeably use optimization of $\mathcal{D}_{\mathsf{syn}}$ *or* $(\mathbf{X}_{\mathsf{syn}}, \mathbf{Y}_{\mathsf{syn}})$ henceforth.

## 2.1 Data Distillation by Meta-model Matching

Meta-model matching-based data distillation approaches fundamentally optimize for the transferability of models trained on the data summary when generalized to the original dataset:

$$\underset{\mathcal{D}_{\mathsf{syn}}}{\arg\min} \quad \mathcal{L}_{\mathcal{D}}\left(\theta^{\mathcal{D}_{\mathsf{syn}}}\right) \quad \text{s.t.} \quad \theta^{\mathcal{D}_{\mathsf{syn}}} \triangleq \underset{\theta}{\arg\min} \; \mathcal{L}_{\mathcal{D}_{\mathsf{syn}}}(\theta), \tag{3}$$

where intuitively, the inner-loop trains a representative learning algorithm on the data summary *until convergence*, and the outer-loop subsequently optimizes the data summary for the transferability of the optimized learning algorithm to the original dataset. Besides common assumptions mentioned earlier, the key simplifying assumption for this family of methods is that a perfect classifier exists and can be estimated on $\mathcal{D}$, *i.e.*, $\exists \; \theta^{\mathcal{D}}$ s.t. $l(\Phi_{\theta^{\mathcal{D}}}(x), y) = 0, \; \forall x \sim \mathcal{X}, y \sim \mathcal{Y}$. Plugging the second assumption along with the `iid` assumption of $\mathcal{D}$ in Equation (2) directly translates to Equation (3). Despite the assumption, Equation (3) is highly expensive both in terms of computation time and memory, due to which, methods from this family typically resort to making further assumptions.

Wang et al. (2018) (DD) originally proposed the task of data distillation, and used the meta-model matching framework for optimization. DD makes the optimization in Equation (3) tractable by performing (1) local optimization *à la* stochastic gradient descent (SGD) in the inner-loop, and (2) outer-loop optimization using Truncated Back-Propagation Through Time (TBPTT), *i.e.*, unrolling only a limited number of inner-loop optimization steps. Formally, the modified optimization objective for DD is as follows:

$$\underset{\mathcal{D}_{\mathsf{syn}}}{\arg\min} \quad \underset{\theta_0 \sim \mathbf{P}_\theta}{\mathbb{E}} \left[ \mathcal{L}_{\mathcal{D}}\left(\theta_T\right) \right] \quad \text{s.t.} \quad \theta_{t+1} \leftarrow \theta_t - \eta \cdot \nabla_\theta \mathcal{L}_{\mathcal{D}_{\mathsf{syn}}}(\theta_t), \tag{4}$$

where $\mathbf{P}_\theta$ is a parameter initialization distribution of choice, $T$ accounts for the truncation in TBPTT, and $\eta$ is a tunable learning rate. We also elucidate DD's control-flow in Algorithm 1 for reference.

Notably, TBPTT has been associated with drawbacks such as (1) computationally expensive inner-loop unrolling; (2) bias involved with truncated unrolling (Wu et al., 2018); and (3) poorly conditioned loss landscapes, particularly with long unrolls (Metz et al., 2019). Consequently, the TBPTT framework was empirically shown to be ineffective for data distillation (Zhao et al., 2021). However, recent work (Deng & Russakovsky, 2022) claims that using momentum-based optimizers and longer inner-loop unrolling can greatly improve performance. We delay a deeper discussion of this work to Section 2.5 for clarity.

---

**Algorithm 1:** Control-flow of data distillation using naïve meta-matching (Equation (4))

---

**Input:** Target dataset $\mathcal{D}$, outer-loop iterations $K$, parameter initialization distribution $\mathbf{P}_\theta$, inner-loop iterations $T$, inner-loop learning rate $\eta$, outer-loop learning rate $\eta_{\mathsf{syn}}$

1 **Initialize:** $(\mathbf{X}_{\mathsf{syn}}^0, \mathbf{Y}_{\mathsf{syn}}^0) \sim \mathcal{D}$

2 **for** $k = 1, \ldots, K$ **do**                                        // Outer-loop: optimize $\mathcal{D}_{\mathsf{syn}}$

3 $\quad$ Initialize $\theta_0 \sim \mathbf{P}_\theta$

4 $\quad$ **for** $t = 1, \ldots, T$ **do**                                 // Inner-loop: optimize $\Phi$ on $\mathcal{D}_{\mathsf{syn}}^{k-1}$

5 $\quad\quad$ $\theta_t \leftarrow \theta_{t-1} - \eta \cdot \nabla_\theta \mathcal{L}_{\mathcal{D}_{\mathsf{syn}}^{k-1}}(\theta_{t-1})$

6 $\quad$ $\mathbf{X}_{\mathsf{syn}}^k \leftarrow \mathbf{X}_{\mathsf{syn}}^{k-1} - \eta_{\mathsf{syn}} \cdot \nabla_{\mathbf{X}_{\mathsf{syn}}} \mathcal{L}_\mathcal{D}(\theta_T)$   // Update $\mathbf{X}_{\mathsf{syn}}$ by computing unrolled meta-gradient

7 $\quad$ $\mathbf{Y}_{\mathsf{syn}}^k \leftarrow \mathbf{Y}_{\mathsf{syn}}^{k-1} - \eta_{\mathsf{syn}} \cdot \nabla_{\mathbf{Y}_{\mathsf{syn}}} \mathcal{L}_\mathcal{D}(\theta_T)$   // Update $\mathbf{Y}_{\mathsf{syn}}$ by computing unrolled meta-gradient

$\quad$ **Output:** $\mathcal{D}_{\mathsf{syn}}^K \equiv (\mathbf{X}_{\mathsf{syn}}^K, \mathbf{Y}_{\mathsf{syn}}^K)$

---

Analogously, a separate line of work focuses on using Neural Tangent Kernel (NTK) (Jacot et al., 2018) based algorithms to solve the inner-loop in closed form. As a brief side note, the infinite-width correspondence states that performing Kernelized Ridge Regression (KRR) using the NTK of a given neural network, is equivalent to training the same $\infty$-width neural network with L2 reconstruction loss for $\infty$ SGD-steps. These "$\infty$-width" neural networks have been shown to perform reasonably compared to their finite-width counterparts, while also being solved in closed-form (see Lee et al. (2020) for a detailed analysis on finite *vs.* infinite neural networks for image classification). KIP uses the NTK of a fully-connected neural network (Nguyen et al., 2021a), or a convolutional network (Nguyen et al., 2021b) in the inner-loop of Equation (3) for efficient data distillation. More formally, given the NTK $\mathcal{K} : \mathcal{X} \times \mathcal{X} \mapsto \mathbb{R}$ of a neural network architecture, KIP optimizes the following objective:

$$\underset{\mathbf{X}_{\mathsf{syn}}, \mathbf{Y}_{\mathsf{syn}}}{\arg\min} \quad \left\| \mathbf{Y} - \mathbf{K}_{\mathbf{X}\mathbf{X}_{\mathsf{syn}}} \cdot (\mathbf{K}_{\mathbf{X}_{\mathsf{syn}}\mathbf{X}_{\mathsf{syn}}} + \lambda I)^{-1} \cdot \mathbf{Y}_{\mathsf{syn}} \right\|^2, \tag{5}$$

where $\mathbf{K}_{AB} \in \mathbb{R}^{|A| \times |B|}$ represents the gramian matrix of two sets $A$ and $B$, and whose $(i, j)^{\text{th}}$ element is defined by $\mathcal{K}(A_i, B_j)$. Although KIP doesn't impose any additional simplifications to the meta-model matching framework, it has an $\mathcal{O}(|\mathcal{D}| \cdot n \cdot \dim(\mathcal{X}))$ time and memory complexity, limiting its scalability. Subsequently, RFAD (Loo et al., 2022) proposes using (1) the light-weight Empirical Neural Network Gaussian Process (NNGP) kernel (Neal, 2012) instead of the NTK; and (2) a classification loss (*e.g.*, NLL) instead of the L2-reconstruction loss for the outer-loop to get $\mathcal{O}(n)$ time complexity while also having better performance. On a similar note, FRePO (Zhou et al., 2022b) decouples the feature extractor and a linear classifier in $\Phi$, and alternatively optimizes (1) the data summary along with the classifier, and (2) the feature extractor. To be precise, let $f_\theta : \mathcal{X} \mapsto \mathcal{X}'$ be the feature extractor, $g_\psi : \mathcal{X}' \mapsto \mathcal{Y}$ be the linear classifier, s.t. $\Phi(x) \equiv g_\psi(f_\theta(x)) \; \forall x \in \mathcal{X}$; the optimization objective for FRePO can be written as:

$$\underset{\mathbf{X}_{\mathsf{syn}}, \mathbf{Y}_{\mathsf{syn}}}{\arg\min} \quad \underset{\theta_0 \sim \mathbf{P}_\theta}{\mathbb{E}} \left[ \sum_{t=0}^{T} \left\| \mathbf{Y} - \mathbf{K}_{\mathbf{X}\mathbf{X}_{\mathsf{syn}}}^{\theta_t} \cdot (\mathbf{K}_{\mathbf{X}_{\mathsf{syn}}\mathbf{X}_{\mathsf{syn}}}^{\theta_t} + \lambda I)^{-1} \cdot \mathbf{Y}_{\mathsf{syn}} \right\|^2 \right]$$
$$\text{s.t.} \quad \theta_{t+1} \leftarrow \theta_t - \eta \cdot \underset{(x,y) \sim \mathcal{D}_{\mathsf{syn}}}{\mathbb{E}} [\nabla_\theta l(g_\psi(f_\theta(x)), y)] \; ; \; \mathbf{K}_{\mathbf{X}_{\mathsf{syn}}\mathbf{X}_{\mathsf{syn}}}^{\theta} \triangleq f_{\theta_t}(\mathbf{X}_{\mathsf{syn}}) f_{\theta_t}(\mathbf{X}_{\mathsf{syn}})^T, \tag{6}$$

where $T$ represents the number of inner-loop update steps for the feature extractor $f_\theta$. Notably, (1) a wide architecture for $f_\theta$ is crucial for distillation quality in FRePO; and (2) despite the bilevel optimization, FRePO is shown to be more scalable compared to KIP (Equation (5)), while also being more generalizable.

## 2.2 Data Distillation by Gradient Matching

Gradient matching based data distillation, at a high level, performs one-step distance matching on (1) the network trained on the target dataset ($\mathcal{D}$) *vs.* (2) the same network trained on the data summary ($\mathcal{D}_{\mathsf{syn}}$). In contrast to the meta-model matching framework, such an approach circumvents the unrolling of the inner-loop, thereby making the overall optimization much more efficient. First proposed by Zhao et al. (2021) (DC),

data summaries optimized by gradient-matching significantly outperformed data pruning methodologies, as well as TBPTT-based data distillation proposed by Wang et al. (2018). Formally, given a learning algorithm $\Phi$, DC solves the following optimization objective:

$$\underset{\mathcal{D}_{\text{syn}}}{\arg\min} \quad \underset{\substack{\theta_0 \sim \mathbf{P}_\theta \\ c \sim \mathcal{C}}}{\mathbb{E}} \left[ \sum_{t=0}^{T} \mathbf{D}\left( \nabla_\theta \mathcal{L}_{\mathcal{D}^c}(\theta_t), \nabla_\theta \mathcal{L}_{\mathcal{D}_{\text{syn}}^c}(\theta_t) \right) \right] \quad \text{s.t.} \quad \theta_{t+1} \leftarrow \theta_t - \eta \cdot \nabla_\theta \mathcal{L}_{\mathcal{D}_{\text{syn}}}(\theta_t), \tag{7}$$

where $T$ accounts for model similarity $T$-steps in the future, and $\mathbf{D} : \mathbb{R}^{|\theta|} \times \mathbb{R}^{|\theta|} \mapsto \mathbb{R}$ is a distance metric of choice (typically cosine distance). In addition to assumptions imposed by the meta-model matching framework (Section 2.1), gradient-matching assumes (1) inner-loop optimization of only $T$ steps; (2) local smoothness: two sets of model parameters close to each other (given a distance metric) imply model similarity; and (3) first-order approximation of $\theta_t^{\mathcal{D}}$: instead of exactly computing the training trajectory of optimizing $\theta_0$ on $\mathcal{D}$ (say $\theta_t^{\mathcal{D}}$); perform first-order approximation on the optimization trajectory of $\theta_0$ on the much smaller $\mathcal{D}_{\text{syn}}$ (say $\theta_t^{\mathcal{D}_{\text{syn}}}$), *i.e.*, approximate $\theta_t^{\mathcal{D}}$ as a single gradient-descent update on $\theta_{t-1}^{\mathcal{D}_{\text{syn}}}$ using $\mathcal{D}$ rather than $\theta_{t-1}^{\mathcal{D}}$ (Figure 3).

Subsequently, numerous other approaches have been built atop this framework with subtle variations. DSA (Zhao & Bilen, 2021) improves over DC by performing the same image-augmentations (*e.g.*, crop, rotate, jitter, *etc.*) on both $\mathcal{D}$ and $\mathcal{D}_{\text{syn}}$ while optimizing Equation (7). Since these augmentations are universal and are applicable across data distillation frameworks, augmentations performed by DSA have become a common part of all methods proposed henceforth, but we omit them for notational clarity. DCC (Lee et al., 2022b) further modifies the gradient-matching objective to incorporate class contrastive signals inside each gradient-matching step and is shown to improve stability as well as performance. With $\theta_t$ evolving similarly as in Equation (7), the modified optimization objective for DCC can be written as:

$$\underset{\mathcal{D}_{\text{syn}}}{\arg\min} \quad \underset{\theta_0 \sim \mathbf{P}_\theta}{\mathbb{E}} \left[ \sum_{t=0}^{T} \mathbf{D}\left( \underset{c \in \mathcal{C}}{\mathbb{E}}\left[ \nabla_\theta \mathcal{L}_{\mathcal{D}^c}(\theta_t) \right], \underset{c \in \mathcal{C}}{\mathbb{E}}\left[ \nabla_\theta \mathcal{L}_{\mathcal{D}_{\text{syn}}^c}(\theta_t) \right] \right) \right] \tag{8}$$

Most recently, Kim et al. (2022) (IDC) extend the gradient matching framework by: (1) multi-formation: to synthesize a higher amount of data within the same memory budget, store the data summary (*e.g.*, images) in a lower resolution to remove spatial redundancies, and upsample (using *e.g.*, bilinear, FSRCNN (Dong et al., 2016)) to the original scale while usage; and (2) matching gradients of the network's training trajectory over the full dataset $\mathcal{D}$ rather than the data summary $\mathcal{D}_{\text{syn}}$. To be specific, given a $k\times$ upscaling function $f : \mathbb{R}^{d \times d} \mapsto \mathbb{R}^{kd \times kd}$, the modified optimization objective for IDC can be formalized as:

$$\underset{\mathcal{D}_{\text{syn}}}{\arg\min} \quad \underset{\substack{\theta_0 \sim \mathbf{P}_\theta \\ c \sim \mathcal{C}}}{\mathbb{E}} \left[ \sum_{t=0}^{T} \mathbf{D}\left( \nabla_\theta \mathcal{L}_{\mathcal{D}^c}(\theta_t), \nabla_\theta \mathcal{L}_{f(\mathcal{D}_{\text{syn}}^c)}(\theta_t) \right) \right] \quad \text{s.t.} \quad \theta_{t+1} \leftarrow \theta_t - \eta \cdot \nabla_\theta \mathcal{L}_{\mathcal{D}}(\theta_t) \tag{9}$$

Kim et al. (2022) further hypothesize that training models on $\mathcal{D}_{\text{syn}}$ instead of $\mathcal{D}$ in the inner-loop has two major drawbacks: (1) strong coupling of the inner- and outer-loop resulting in a chicken-egg problem (McLachlan & Krishnan, 2007); and (2) vanishing network gradients due to the small size of $\mathcal{D}_{\text{syn}}$, leading to an improper outer-loop optimization for gradient-matching based techniques.

## 2.3 Data Distillation by Trajectory Matching

Cazenavette et al. (2022) proposed MTT which aims to match the training trajectories of models trained on $\mathcal{D}$ *vs.* $\mathcal{D}_{\text{syn}}$. More specifically, let $\{\theta_t^{\mathcal{D}}\}_{t=0}^{T}$ represent the training trajectory of training $\Phi_\theta$ on $\mathcal{D}$; trajectory matching algorithms aim to solve the following optimization:

$$\underset{\mathcal{D}_{\text{syn}}, \eta}{\arg\min} \quad \underset{\theta_0 \sim \mathbf{P}_\theta}{\mathbb{E}} \left[ \sum_{t=0}^{T-M} \frac{\mathbf{D}\left( \theta_{t+M}^{\mathcal{D}}, \theta_{t+N}^{\mathcal{D}_{\text{syn}}} \right)}{\mathbf{D}\left( \theta_{t+M}^{\mathcal{D}}, \theta_t^{\mathcal{D}} \right)} \right] \tag{10}$$

$$\text{s.t.} \quad \theta_{t+i+1}^{\mathcal{D}_{\text{syn}}} \leftarrow \theta_{t+i}^{\mathcal{D}_{\text{syn}}} - \eta \cdot \nabla_\theta \mathcal{L}_{\mathcal{D}_{\text{syn}}}(\theta_{t+i}^{\mathcal{D}_{\text{syn}}}) \quad ; \quad \theta_{t+1}^{\mathcal{D}_{\text{syn}}} \leftarrow \theta_t^{\mathcal{D}} - \eta \cdot \nabla_\theta \mathcal{L}_{\mathcal{D}_{\text{syn}}}(\theta_t^{\mathcal{D}}),$$

where $\mathbf{D} : \mathbb{R}^{|\theta|} \times \mathbb{R}^{|\theta|} \mapsto \mathbb{R}$ is a distance metric of choice (typically L2 distance). Such an optimization can intuitively be seen as optimizing for similar quality models trained with $N$ SGD steps on $\mathcal{D}_{\text{syn}}$, compared to

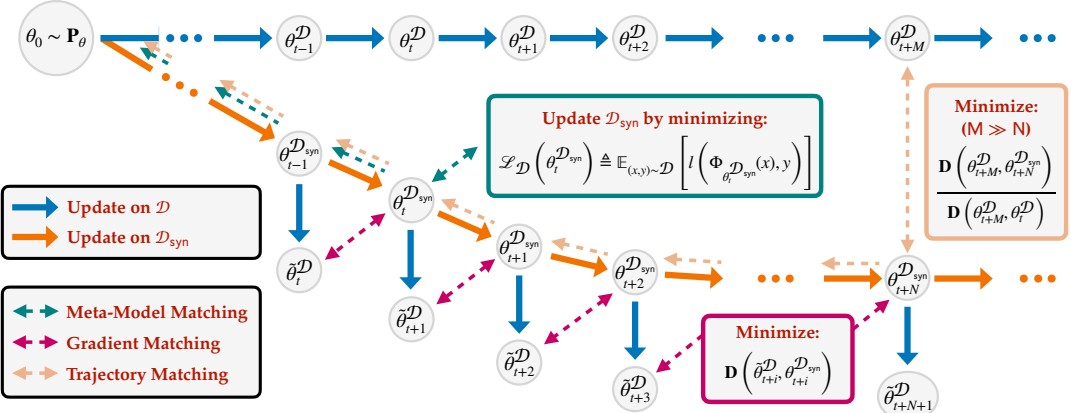

[HQ Image Link] Figure 3: The underlying optimization in various data distillation frameworks.

$M \gg N$ steps on $\mathcal{D}$, thereby invoking long-horizon trajectory matching. Notably, calculating the gradient of Equation (10) *w.r.t.* $\mathcal{D}_{\text{syn}}$ encompasses gradient unrolling through $N$-timesteps, thereby limiting the scalability of MTT. On the other hand, since the trajectory of training $\Phi_\theta$ on $\mathcal{D}$, *i.e.*, $\{\theta_t^{\mathcal{D}}\}_{t=0}^T$ is independent of the optimization of $\mathcal{D}_{\text{syn}}$, it can be pre-computed for various $\theta_0 \sim \mathbf{P}_\theta$ initializations and directly substituted. Similar to gradient matching methods (Section 2.2), the trajectory matching framework also optimizes the first-order distance between parameters, thereby inheriting the local smoothness assumption. As a scalable alternative, Cui et al. (2022b) proposed TESLA, which re-parameterizes the parameter-matching loss of MTT in Equation (10) (specifically when $\mathbf{D}$ is set as the L2 distance), using linear algebraic manipulations to make the bilevel optimization's memory complexity independent of $N$. Furthermore, TESLA uses learnable soft-labels ($\mathbf{Y}_{\text{syn}}$) during the optimization for an increased compression efficiency.

## 2.4 Data Distillation by Distribution Matching

Even though the aforementioned gradient-matching or trajectory-matching based data distillation techniques have been empirically shown to synthesize high-quality data summaries, the underlying bilevel optimization, however, is oftentimes a computationally expensive procedure. To this end, distribution-matching techniques solve a correlated proxy task via a single-level optimization, leading to a vastly improved scalability. More specifically, instead of matching the quality of models on $\mathcal{D}$ *vs.* $\mathcal{D}_{\text{syn}}$, distribution-matching techniques directly match the distribution of $\mathcal{D}$ *vs.* $\mathcal{D}_{\text{syn}}$. The key assumption for this family of methods is that two datasets that are similar according to a particular distribution divergence metric, lead to similarly trained models.

First proposed by Zhao & Bilen (2023), DM uses (1) numerous parametric encoders to cast high-dimensional data into respective low-dimensional latent spaces; and (2) an approximation of the Maximum Mean Discrepancy to compute the distribution mismatch between $\mathcal{D}$ and $\mathcal{D}_{\text{syn}}$ in each of the latent spaces. More precisely, given a set of $k$ encoders $\mathcal{E} \triangleq \{\psi_i : \mathcal{X} \mapsto \mathcal{X}_i\}_{i=1}^k$, the optimization objective can be written as:

$$\underset{\mathcal{D}_{\text{syn}}}{\arg\min} \quad \underset{\substack{\psi \sim \mathcal{E} \\ c \sim \mathcal{C}}}{\mathbb{E}} \left[ \left\| \underset{x \sim \mathcal{D}^c}{\mathbb{E}} \left[ \psi(x) \right] - \underset{x \sim \mathcal{D}_{\text{syn}}^c}{\mathbb{E}} \left[ \psi(x) \right] \right\|^2 \right] \tag{11}$$

DM uses a set of randomly initialized neural networks (with the same architecture) to instantiate $\mathcal{E}$. They observe similar performance when instantiated with more meaningful, task-optimized neural networks, despite it being much less efficient. CAFE (Wang et al., 2022) further refines the distribution-matching idea by: (1) solving a bilevel optimization problem for jointly optimizing a *single* encoder ($\Phi$) and the data summary, rather than using a pre-determined *set* of encoders ($\mathcal{E}$); and (2) assuming a neural network encoder ($\Phi$), match the latent representations obtained at all intermediate layers of the encoder instead of only the last layer. Formally, given a $(L+1)$-layer neural network $\Phi_\theta : \mathcal{X} \mapsto \mathcal{Y}$ where $\Phi_\theta^l$ represents $\Phi$'s output at the $l^{\text{th}}$

layer, the optimization problem for CAFE can be specified as:

$$\underset{\mathcal{D}_{\mathsf{syn}}}{\arg\min} \quad \underset{c \sim \mathcal{C}}{\mathbb{E}} \left[ \sum_{l=1}^{L} \left\| \underset{x\sim\mathcal{D}^c}{\mathbb{E}} \left[ \Phi_{\theta_t}^l(x) \right] - \underset{x\sim\mathcal{D}_{\mathsf{syn}}^c}{\mathbb{E}} \left[ \Phi_{\theta_t}^l(x) \right] \right\|^2 - \beta \cdot \underset{(x,y)\sim\mathcal{D}^c}{\mathbb{E}} \left[ \log \hat{p}(y|x,\theta_t) \right] \right]$$

$$\text{s.t.} \quad \theta_{t+1} \leftarrow \theta_t - \eta \cdot \nabla_\theta \mathcal{L}_{\mathcal{D}_{\mathsf{syn}}}(\theta_t) \;\; ; \;\; \hat{p}(y|x,\theta) \triangleq \underset{y}{\text{softmax}} \left( \left\langle \Phi_\theta^L(x), \underset{x'\sim\mathcal{D}_{\mathsf{syn}}^y}{\mathbb{E}} \left[ \Phi_\theta^L(x') \right] \right\rangle \right),$$

(12)

where $\hat{p}(\cdot|\cdot,\theta)$ intuitively represents the nearest centroid classifier on $\mathcal{D}_{\mathsf{syn}}$ using the latent representations obtained by last layer of $\Phi_\theta$. Analogously, IT-GAN (Zhao & Bilen, 2022) also uses the distribution-matching framework in Equation (11) to generate data that is informative for model training, in contrast to the traditional GAN (Goodfellow et al., 2014) which focuses on generating realistic data.

## 2.5 Data Distillation by Factorization

All of the aforementioned data distillation frameworks intrinsically maintain the synthesized data summary as a large set of free parameters, which are in turn optimized. Arguably, such a setup prohibits knowledge sharing between synthesized data points (parameters), which might introduce data redundancy. On the other hand, factorization-based data distillation techniques parameterize the data summary using two separate components: (1) bases: a set of mutually independent base vectors; and (2) hallucinators: a mapping from the bases' vector space to the joint data- and label-space. In turn, both the bases and hallucinators are optimized for the task of data distillation.

Formally, let $\mathcal{B} \triangleq \{b_i \in \mathbb{B}\}_{i=1}^{|\mathcal{B}|}$ be the set of bases, and $\mathcal{H} \triangleq \{h_i : \mathbb{B} \mapsto \mathcal{X} \times \mathcal{Y}\}_{i=1}^{|\mathcal{H}|}$ be the set of hallucinators, then the data summary is parameterized as $\mathcal{D}_{\mathsf{syn}} \triangleq \{h(b)\}_{b\sim\mathcal{B},\ h\sim\mathcal{H}}$. Even though such a two-pronged approach seems similar to generative modeling of data, note that unlike classic generative models, (1) the input space consists *only of* a fixed and optimized set of latent codes and isn't meant to take any other inputs; and (2) given a specific $\mathcal{B}$ and $\mathcal{H}$, we can generate at most $|\mathcal{B}| \cdot |\mathcal{H}|$ sized data summaries. Notably, such a hallucinator-bases data parameterization can be optimized using any of the aforementioned data optimization frameworks (Sections 2.1 to 2.4)

This framework was concurrently proposed by Deng & Russakovsky (2022) (we take the liberty to term their unnamed model as "*Lin*-ear *Ba*-ses") and Liu et al. (2022c) (HaBa). LinBa modifies the general hallucinator-bases framework by assuming (1) the bases' vector space ($\mathbb{B}$) to be the same as the task input space ($\mathcal{X}$); and (2) the hallucinator to be linear and additionally conditioned on a given predictand. More specifically, the data parameterization can be formalized as follows:

$$\mathcal{D}_{\mathsf{syn}} \triangleq \left\{ (y\, \mathbf{H}^T\mathbf{B},\ y) \right\}_{\substack{y\sim\mathcal{C} \\ \mathbf{H}\sim\mathcal{H}}}$$

$$\text{s.t.} \quad \mathbf{B} \in \mathbb{R}^{|\mathbf{B}|\times\dim(\mathcal{X})} \triangleq [b_i \in \mathcal{X}]_{i=1}^{|\mathbf{B}|} \quad ; \quad \mathcal{H} \triangleq \left\{ \mathbf{H}_i \in \mathbb{R}^{|\mathbf{B}|\times|\mathcal{C}|} \right\}_{i=1}^{|\mathcal{H}|},$$

(13)

where for the sake of notational simplicity, we assume $y \in \mathbb{R}^{|\mathcal{C}|}$ represents the one-hot vector of the label for which we want to generate data, and the maximum amount of data that can be synthesized $n \leq |\mathcal{C}| \cdot |\mathcal{H}|$. Since the data generation (Equation (13)) is end-to-end differentiable, both $\mathbf{B}$ and $\mathcal{H}$ are jointly optimized using the TBPTT framework discussed in Section 2.1, albeit with some crucial modifications for vastly improved performance: (1) using momentum-based optimizers instead of vanilla SGD in the inner-loop; and (2) longer unrolling ($\geq 100$ steps) of the inner-loop during TBPTT. Liu et al. (2022c) (HaBa) relax the linear and predictand-conditional hallucinator assumption of LinBa, equating to the following data parameterization:

$$\mathcal{D}_{\mathsf{syn}} \triangleq \left\{ (h(b),\ y) \right\}_{\substack{b,y\sim\mathcal{B} \\ h\sim\mathcal{H}}} \quad \text{s.t.} \quad \mathcal{B} \triangleq \left\{ (b_i \in \mathcal{X}, y_i \in \mathcal{Y}) \right\}_{i=1}^{|\mathcal{B}|} \quad ; \quad \mathcal{H} \triangleq \{h_{\theta_i} : \mathcal{X} \mapsto \mathcal{X}\}_{i=1}^{|\mathcal{H}|},$$

(14)

where $\mathcal{B}$ and $\mathcal{H}$ are optimized using the trajectory matching framework (Section 2.3) with an additional contrastive constraint to promote diversity in $\mathcal{D}_{\mathsf{syn}}$ (cf. Liu et al. (2022c), Equation (6)). Following this setup, HaBa can generate at most $|\mathcal{B}| \cdot |\mathcal{H}|$ sized data summaries. Furthermore, one striking difference between HaBa (Equation (14)) and LinBa (Equation (13)) is that to generate each data point, LinBa uses a linear combination of *all* the bases, whereas HaBa generates a data point using a *single* base vector.

Table 1: Comparison of data distillation methods. Each method (1) synthesizes the data summary on the train-set; (2) unless mentioned, trains a 128-width ConvNet (Gidaris & Komodakis, 2018) on the data summary; and (3) evaluates it on the test-set. Confidence intervals are obtained by training at least 5 networks on the data summary. LinBa (No Fact.) represents LinBa with the no factorization. Methods evaluated using KRR are marked as (∞-Conv) or (∞-FC). The equivalent storage-in-bytes is used for factorization-based techniques instead of IPC. The best method in their category is **emboldened**, the best-overall non-factorized method evaluated on ConvNet is colored orange, and the best-overall factorized method is colored blue.

| | Dataset | MNIST | | | CIFAR-10 | | | CIFAR-100 | | | Tiny ImageNet | | |
|---|---|---|---|---|---|---|---|---|---|---|---|---|---|
| | Imgs/Class (IPC) | 1 | 10 | 50 | 1 | 10 | 50 | 1 | 10 | 50 | 1 | 10 | 50 |
| Baselines | Random | 64.9 ±3.5 | **95.1** ±0.9 | **97.9** ±0.2 | 14.4 ±2.0 | 26.0 ±1.2 | 43.4 ±1.0 | 4.2 ±0.3 | 14.6 ±0.5 | 30.0 ±0.4 | **1.5** ±0.1 | 6.0 ±0.8 | **16.8** ±1.8 |
| | Herding[1] | **89.2** ±1.6 | 93.7 ±0.3 | 94.9 ±0.2 | **21.5** ±1.2 | **31.6** ±0.7 | 40.4 ±0.6 | 8.4 ±0.3 | 17.3 ±0.5 | 33.7 ±0.5 | - | - | - |
| | Forgetting[2] | 35.5 ±5.6 | 68.1 ±3.3 | 88.2 ±1.2 | 13.5 ±1.2 | 23.3 ±1.0 | 23.3 ±1.1 | 4.5 ±0.2 | 15.1 ±0.3 | 30.5 ±0.3 | - | - | - |
| Meta-model Matching | DD[3] | - | 79.5 ±8.1 | - | - | 36.8 ±1.2 | - | - | - | - | - | - | - |
| | LinBa (No Fact.)[16] | 95.2 ±0.3 | 98.8 ±0.1 | 99.2 ±0.1 | 49.1 ±0.6 | 62.4 ±0.4 | 70.5 ±0.4 | 21.3 ±0.6 | 34.7 ±0.5 | - | - | - | - |
| | KIP (ConvNet)[4] | 90.1 ±0.1 | 97.5 ±0.0 | 98.3 ±0.1 | 49.9 ±0.2 | 62.7 ±0.3 | 68.6 ±0.2 | 15.7 ±0.2 | 28.3 ±0.1 | - | - | - | - |
| | RFAD (ConvNet)[5] | 94.4 ±1.5 | 98.5 ±0.1 | 98.8 ±0.1 | 53.6 ±1.2 | 66.3 ±0.5 | 71.1 ±0.4 | 26.3 ±1.1 | 33.0 ±0.3 | - | - | - | - |
| | FRePO (ConvNet)[6] | 93.0 ±0.4 | 98.6 ±0.1 | 99.2 ±0.1 | 46.8 ±0.7 | 65.5 ±0.6 | 71.7 ±0.2 | 28.7 ±0.1 | 42.5 ±0.2 | 44.3 ±0.2 | 15.4 ±0.3 | 25.4 ±0.2 | - |
| | KIP (∞-FC)[7] | 85.5 ±0.1 | 97.2 ±0.2 | 98.4 ±0.1 | 40.5 ±0.4 | 53.1 ±0.5 | 58.6 ±0.4 | - | - | - | - | - | - |
| | KIP (∞-Conv)[4] | 97.3 ±0.1 | 99.1 ±0.1 | 99.5 ±0.1 | 64.7 ±0.2 | 75.6 ±0.2 | 80.6 ±0.1 | 34.9 ±0.1 | 49.5 ±0.3 | - | - | - | - |
| | RFAD (∞-Conv)[5] | 97.2 ±0.2 | 99.1 ±0.0 | 99.1 ±0.0 | 61.4 ±0.8 | 73.7 ±0.2 | 76.6 ±0.3 | **44.1** ±0.1 | 46.8 ±0.2 | - | - | - | - |
| | FRePO (∞-Conv)[6] | 92.6 ±0.4 | 98.6 ±0.1 | 99.2 ±0.1 | 47.9 ±0.6 | 68.0 ±0.2 | 74.4 ±0.1 | 32.3 ±0.1 | 44.9 ±0.2 | 43.0 ±0.3 | 19.1 ±0.3 | 26.5 ±0.1 | - |
| Gradient Matching | DC[8] | **91.7** ±0.5 | 97.4 ±0.2 | 98.2 ±0.2 | 28.3 ±0.5 | 44.9 ±0.5 | 53.9 ±0.5 | 12.8 ±0.3 | 25.2 ±0.3 | 30.5 ±0.3 | 4.6 ±0.6 | 11.2 ±1.6 | 10.9 ±0.7 |
| | DSA[9] | 88.7 ±0.6 | **97.8** ±0.1 | **99.2** ±0.1 | 28.8 ±0.7 | 52.1 ±0.5 | 60.6 ±0.5 | 13.9 ±0.3 | 32.3 ±0.3 | **42.8** ±0.4 | 6.6 ±0.2 | 14.4 ±2.0 | 22.6 ±2.6 |
| | DCC[10] | - | - | - | 34.0 ±0.7 | 54.5 ±0.5 | 64.2 ±0.4 | 14.6 ±0.3 | 33.5 ±0.3 | 39.3 ±0.4 | - | - | - |
| Distr. Matching | DM[11] | 89.7 ±0.6 | **97.5** ±0.1 | 98.6 ±0.1 | 26.0 ±0.8 | 48.9 ±0.6 | 63.0 ±0.4 | 11.4 ±0.3 | 29.7 ±0.3 | 43.6 ±0.4 | 3.9 ±0.2 | 12.9 ±0.4 | 24.1 ±0.3 |
| | CAFE[12] | 90.8 ±0.5 | 97.5 ±0.1 | 98.9 ±0.2 | 31.6 ±0.8 | 50.9 ±0.5 | 62.3 ±0.4 | 14.0 ±0.3 | 31.5 ±0.2 | 42.9 ±0.2 | - | - | - |
| Traj. Matching | MTT[13] | - | - | - | 46.3 ±0.8 | 65.3 ±0.7 | 71.6 ±0.2 | 24.3 ±0.3 | 40.1 ±0.4 | 47.7 ±0.2 | 8.8 ±0.3 | 23.2 ±0.2 | 28.0 ±0.3 |
| | TESLA[14] | - | - | - | 48.5 ±0.8 | 66.4 ±0.8 | 72.6 ±0.7 | 24.8 ±0.4 | 41.7 ±0.3 | 47.9 ±0.3 | - | - | - |
| Factorization | IDC[15] | - | - | - | 50.0 ±0.4 | 67.5 ±0.5 | 74.5 ±0.1 | - | 44.8 ±0.2 | - | - | - | - |
| | LinBa[16] | **98.7** ±0.7 | **99.3** ±0.5 | **99.4** ±0.4 | **66.4** ±0.4 | 71.2 ±0.4 | 73.6 ±0.5 | 34.0 ±0.4 | 42.9 ±0.7 | - | 16.0 ±0.7 | - | - |
| | HaBa[17] | - | - | - | 48.3 ±0.8 | 69.9 ±0.4 | 74.0 ±0.2 | 33.4 ±0.4 | 40.2 ±0.2 | **47.0** ±0.2 | - | - | - |
| | KFS[18] | - | - | - | 59.8 ±0.5 | **72.0** ±0.3 | **75.0** ±0.2 | **40.0** ±0.5 | **50.6** ±0.2 | - | **22.7** ±0.2 | **27.8** ±0.2 | - |
| Full Dataset | | | 99.6 ±0.1 | | | 84.8 ±0.1 | | | 56.2 ±0.3 | | | 37.6 ±0.4 | |

[1] (Welling, 2009), [2] (Toneva et al., 2019), [3] (Wang et al., 2018), [4] (Nguyen et al., 2021b), [5] (Loo et al., 2022)

[6] (Zhou et al., 2022b), [7] (Nguyen et al., 2021a), [8] (Zhao et al., 2021), [9] (Zhao & Bilen, 2021), [10] (Lee et al., 2022b)

[11] (Zhao & Bilen, 2023), [12] (Wang et al., 2022), [13] (Cazenavette et al., 2022), [14] (Cui et al., 2022b)

[15] (Kim et al., 2022), [16] (Deng & Russakovsky, 2022), [17] (Liu et al., 2022c), [18] (Lee et al., 2022a)

Lee et al. (2022a) (KFS) further build atop this framework by maintaining a different bases' vector space $\mathbb{B}$ from the data domain $\mathcal{X}$, such that $\dim(\mathbb{B}) < \dim(\mathcal{X})$. This parameterization allows KFS to store an even larger number of images, with a comparable storage budget to other methods. Formally, the data parameterization for KFS can be specified as:

$$
\mathcal{D}_{\text{syn}} \triangleq \bigcup_{c \in \mathcal{C}} \{ (h(b),\ c) \}_{\substack{b \sim \mathcal{B}_c \\ h \sim \mathcal{H}}}
$$

$$
\text{s.t.} \quad \mathcal{B} \triangleq \bigcup_{c \in \mathcal{C}} \mathcal{B}_c \ ; \quad \mathcal{B}_c \triangleq \{ b_i^c \in \mathbb{B} \}_{i=1}^B \quad ; \quad \mathcal{H} \triangleq \{ h_{\theta_i} : \mathbb{B} \mapsto \mathcal{X} \}_{i=1}^{|\mathcal{H}|} ,
\tag{15}
$$

where KFS stores $B$ bases per class, equivalent to a total of $n = |\mathcal{C}| \cdot B \cdot |\mathcal{H}|$ sized data summaries. Following this data parameterization, $\mathcal{B}$ and $\mathcal{H}$ are optimized using the distribution matching framework for data distillation (Equation (11)) to ensure fast, single-level optimization.

**Data Distillation *vs.* Data Compression.** We highlight that it is non-trivial to ensure a fair comparison between data distillation techniques that (1) are "non-factorized", *i.e.*, maintain each synthesized data point as a set of free-parameters (Sections 2.1 to 2.4); and (2) use factorized approaches discussed in this section to efficiently organize the data summary. If we use the size of the data summary ($n$) as the efficiency metric, factorized approaches are adversely affected as they need a much smaller storage budget to synthesize the same-sized data summaries. On the other hand, if we use "end-to-end bytes of storage" as the efficiency metric, non-factorized approaches are adversely affected as they perform no kind of data compression, but focus solely on better understanding the model-to-data relationship through the lens of optimization. For a better intuition, one can apply posthoc lossless compression (*e.g.*, Huffman coding) on data synthesized by non-factorized data distillation approaches to fit more images in the same storage budget (Schirrmeister et al., 2022). Such techniques unintentionally deviate from the original intent of data distillation, and progress more toward better data compression techniques. As a potential solution, we encourage the community to consider reporting results for both scenarios: a fixed data summary size $n$, as well as fixed bytes-of-storage. Nonetheless, for the ease of empirical comparison amongst the discussed data distillation techniques, we provide a collated set of results over four image-classification datasets in Table 1.

## 3    Data Modalities

Having learned about different kinds of optimization frameworks for data distillation, we now discuss an orthogonal (and important) aspect of data distillation — *what kinds of data can data distillation techniques summarize?* From continuous-valued images to heterogeneous and semi-structured graphs, the underlying data for each unique application of machine learning has its own modality, structure, and set of assumptions. While the earliest data distillation techniques were designed to summarize images, recent steps have been taken to expand the horizon of data distillation into numerous other scenarios. In what follows, we categorize existing data distillation methods by their intended data modality, while also discussing their unique challenges.

**Images.** A large-portion of existing data distillation techniques are designed for image classification data (Cazenavette et al., 2022; Deng & Russakovsky, 2022; Kim et al., 2022; Lee et al., 2022a;b; Liu et al., 2022c; Loo et al., 2022; Nguyen et al., 2021a;b; Wang et al., 2022; 2018; Zhao & Bilen, 2021; 2022; 2023; Zhao et al., 2021; Zhou et al., 2022b) simply because images have a real-valued, continuous data-domain ($\mathcal{X} \equiv \mathbb{R}^{d \times d}$). This allows SGD-based optimization directly on the data, which is treated as a set of free parameters. Intuitively, incrementally changing each pixel value can be treated as slight perturbations in the color space, and hence given a suitable data distillation loss, can be naïvely optimized using SGD.

**Text.** Textual data is available in large amounts from sources like websites, news articles, academic manuscripts, *etc.*, and is also readily accessible with datasets like the common crawl[1] which sizes up to almost 541TB. Furthermore, with the advent of large language models (LLM) (Brown et al., 2020; Devlin et al., 2019; Thoppilan et al., 2022), training such models from scratch on large datasets has become an increasingly

---

[1]https://commoncrawl.org/the-data/

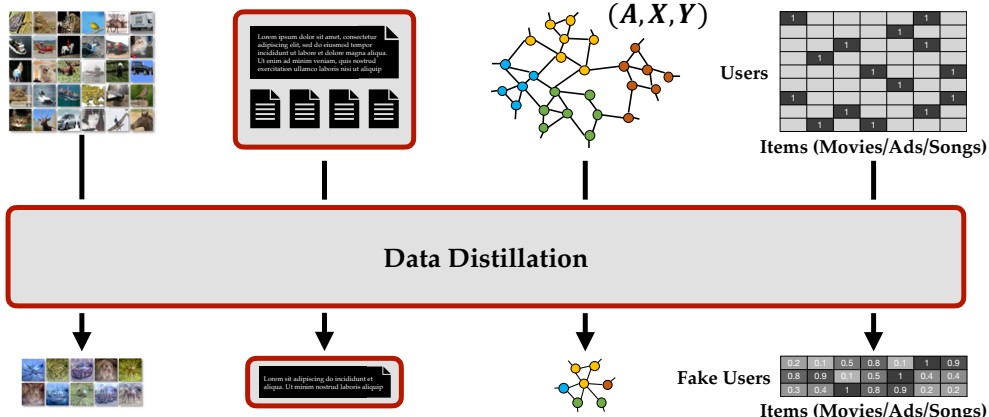

[HQ Image Link] Figure 4: Overview of distilling data for a few commonly observed data modalities.

expensive procedure. Despite recent efforts in democratizing LLM training (Geiping & Goldstein, 2022; Scao et al., 2022; Wolf et al., 2020), effectively distilling large-scale textual data as a solution is yet to be explored. The key bottlenecks for distilling textual data are: (1) the inherently discrete nature of data, where a token should belong in a limited vocabulary of words; (2) the presence of a rich underlying structure, *i.e.*, sentences of words (text) obey fixed patterns according to a grammar; and (3) richness of context, *i.e.*, a given piece of text could have wildly different semantic interpretations under different contexts.

Sucholutsky & Schonlau (2021) take a latent-embedding approach to textual data distillation. On a high level, to circumvent the discreteness of the optimization, the authors perform distillation in a continuous embedding space. More specifically, assuming access to a latent space specified by a *fixed* text-encoder, the authors learn continuous *representations* of each word in the distilled text and optimize it using the TBPTT data-distillation framework proposed by Wang et al. (2018) (Equation (4)). Finally, the distilled text representations are decoded by following a simple nearest-neighbor protocol.

**Graphs.** A wide variety of data and applications can inherently be modeled as graphs, *e.g.*, user-item interactions (Mittal et al., 2021; Sachdeva & McAuley, 2020; Wu et al., 2020), social networks (Fan et al., 2019), autonomous driving (Casas et al., 2020; Sachdeva et al., 2022b), *etc.* Taking the example of social networks, underlying user-user graphs typically size up to the billion-scale (Chen et al., 2021), calling for principled scaling solutions. Graph distillation trivially solves a majority of the scale challenges, but synthesizing tiny, high-fidelity graphs has the following hurdles: (1) nodes in a graph can be highly abstract, *e.g.*, users, products, *etc.* and could be discrete, heterogeneous, or even numerical IDs; (2) graphs follow a variety of intrinsic patterns (*e.g.*, spatial (Kipf & Welling, 2017)) which need to be retained in the distilled graphs; and (3) quadratic size of the adjacency matrix could be computationally prohibitive for data optimization.

Jin et al. (2022b) propose GCOND which distills graphs in the inductive node-classification setting, specified by its node-feature matrix $\mathbf{X}$, adjacency matrix $\mathbf{A}$, and node-target matrix $\mathbf{Y}$. GCOND distills the given graph by learning a synthetic node-feature matrix $\mathbf{X}_{\mathsf{syn}}$, and using $\mathbf{X}_{\mathsf{syn}}$ to generate $\mathbf{A}_{\mathsf{syn}} \triangleq f_\theta(\mathbf{X}_{\mathsf{syn}})$ which can be realized, *e.g.*, through a parametric similarity function $\mathrm{sim}_\theta(\cdot, \cdot)$ between the features of two nodes, *i.e.*, $\mathbf{A}_{\mathsf{syn}}^{i,j} \triangleq \sigma(\mathrm{sim}_\theta(\mathbf{X}_{\mathsf{syn}}^i, \mathbf{X}_{\mathsf{syn}}^j))$, where $\sigma(\cdot)$ is the sigmoid function. Finally, both $\mathbf{X}_{\mathsf{syn}}$ and $\theta$ are optimized using the gradient-matching framework proposed by Zhao et al. (2021) (Equation (7)). Another work (Liu et al., 2022a) (GCDM) shares the same framework as GCOND but instead uses the distribution matching framework proposed by Zhao & Bilen (2023) (Equation (11)) to optimize $\mathbf{X}_{\mathsf{syn}}$ and $\theta$. Extending to a graph-classification setting, Jin et al. (2022a) further propose DOSCOND with two major changes compared to GCOND: (1) instead of parameterizing the adjacency matrix using a similarity function on $\mathbf{X}_{\mathsf{syn}}$, they maintain a free-parameter matrix $\Omega$ with the same size as the adjacency matrix, and sample each $\mathbf{A}_{\mathsf{syn}}^{i,j}$ entry through an independent Bernoulli draw on $\Omega^{i,j}$ as the prior using the reparameterization trick (Maddison et al., 2017). Such a procedure ensures differentiability as well as discrete matrix synthesis; and (2) $\mathbf{X}_{\mathsf{syn}}$ and

$\Omega$ are still optimized using the gradient-matching framework (Equation (7)), albeit with only a single-step, *i.e.*, $T = 1$ for improved scalability and without empirically observing a loss in performance.

**Recommender Systems.** The amount of online user-feedback data available for training recommender systems is rapidly increasing (Wu et al., 2022). Furthermore, typical user-facing recommender systems need to be periodically re-trained (Naumov et al., 2019), which adds to requirements for smarter data summarization solutions (see Sachdeva et al. (2022c) for background on sampling recommender systems data). However, distilling recommender systems data has the following challenges: (1) the data is available in the form of abstract and discrete (`userID`, `itemID`, `relevance`) tuples, which departs from the typical (`features`, `label`) setup; (2) the distribution of both user- and item-popularity follows a strong power-law which leads to data scarcity and unstable optimization; and (3) the data inherits a variety of inherent structures, *e.g.*, sequential patterns (Kang & McAuley, 2018; Sachdeva et al., 2019), user-item graph patterns (Wu et al., 2019), item-item co-occurrence patterns (Steck, 2019), missing-not-at-randomness (Sachdeva et al., 2020; Schnabel et al., 2016), *etc.*

Sachdeva et al. (2022a) propose DISTILL-CF which distills implicit-feedback recommender systems data, *i.e.*, when the observed user-item relevance is binary (*e.g.*, click or no-click). Such data can be visualized as a binary user-item matrix $\mathbf{R}$ where each row represents a single user, and each column represents an item. On a high-level, DISTILL-CF synthesizes fake users along with their item-consumption histories, visualized as a synthetic user-item matrix $\mathbf{R}_{\mathsf{syn}}$. Notably, to preserve semantic meaning, the item-space in $\mathbf{R}_{\mathsf{syn}}$ is the same as in $\mathbf{R}$. To alleviate the data discreteness problem, DISTILL-CF maintains a sampling-prior matrix $\Omega$ which has the same size as $\mathbf{R}_{\mathsf{syn}}$, and can in-turn be used to generate $\mathbf{R}_{\mathsf{syn}}$ using multi-step Gumbel sampling with replacement (Jang et al., 2017) for each user's prior in $\Omega$ (equivalent to each row). Such a formulation automatically also circumvents the dynamic user- and item-popularity artifact in recommender systems data, which can analogously be controlled by the row- and column-wise entropy of $\Omega$. Finally, $\Omega$ is optimized using the meta-model matching framework proposed by Nguyen et al. (2021a). Notably, Sachdeva et al. (2022a) also propose infinite-width autoencoders which suit the task of item recommendation while also leading to closed-form computation of the inner-loop in the meta-model matching framework (Equation (5)).

# 4 Applications

While the data distillation task was originally designed to accelerate model training, there are numerous other applications of a high-fidelity data summary. Below we briefly discuss a few such promising applications, along with providing pointers to existing works.

**Differential Privacy.** Data distillation was recently shown to be a promising solution for differential privacy as defined by Dwork (2008). Dong et al. (2022) show that data distillation techniques can perform better than existing state-of-the-art differentially-private data generators (Cao et al., 2021; Harder et al., 2021) on both performance and privacy grounds. Notably, the privacy benefits of data distillation techniques are virtually *free*, as none of these methods were optimized for generating differentially-private data. Chen et al. (2022) further modify the gradient matching framework (Equation (7)) by clipping and adding white noise to the gradients obtained on the original dataset while optimization. Such a routine was shown to have better sample utility, while also satisfying strict differential privacy guarantees. From a completely application perspective, data distillation has been used to effectively distill sensitive medical data as well (Li et al., 2020a; 2022).

**Neural Architecture Search (NAS).** Automatic searching of neural-network architectures can alleviate a majority of manual effort, as well as lead to more accurate models (see Elsken et al. (2019) for a detailed review). Analogous to using model extrapolation, *i.e.*, extrapolating the performance of an under-trained model architecture on the full dataset; data extrapolation, on the other hand, aims to train models on a small, high-fidelity data sample till convergence. Zhao et al. (2021) show promise of their technique (DC) on a small custom NAS test-bed consisting of only 720 variations of the ConvNet architecture (Gidaris & Komodakis, 2018) by employing the data extrapolation framework. However, Cui et al. (2022a) show that data distillation *does not* perform well when evaluating diverse architectures on the bigger test-bed, NAS-Bench-201 (Dong & Yang, 2020), calling for better rank-preserving data distillation techniques.

**Continual Learning.** Never-ending learning (see Parisi et al. (2019) for a detailed review) has been frequently associated with catastrophic forgetting (French, 1999), *i.e.*, patterns extracted from old data/tasks are easily forgotten when patterns from new data/tasks are learned. Data distillation has been shown as an effective solution to alleviate catastrophic forgetting, by simply using the distilled data summary in a replay buffer that is continually updated and used in subsequent data/task training (Rosasco et al., 2021; Sangermano et al., 2022; Wiewel & Yang, 2021). Deng & Russakovsky (2022) show further evidence of a simple *compress-then-recall* strategy outperforming existing state-of-the-art continual learning approaches. Notably, *only* the data summary is stored for each task, and a new model is trained (from scratch) using all previous data summaries, for each new incoming task.

**Federated Learning.** Federated or collaborative learning (see Li et al. (2020b) for a detailed survey) involves training a learning algorithm in a decentralized fashion. A standard approach to federated learning is to synchronize local parameter updates to a central server, instead of synchronizing the raw data itself (Konečnỳ et al., 2016). Data distillation, on the other hand, alleviates the need to synchronize large parametric models across clients and servers, by synchronizing tiny synthesized data summaries to the central server instead. Subsequently, the entire training happens only on the central server. Such data distillation-based federated learning methods (Goetz & Tewari, 2020; Hu et al., 2022; Liu et al., 2022b; Song et al., 2022; Xiong et al., 2022; Zhou et al., 2020) are shown to perform better than model-synchronization based federated learning approaches, while also requiring multiple orders lesser client-server communication.

## 5 Challenges & Future Directions

Despite achieving remarkable progress in data-efficient learning, there are numerous framework-based, theoretical, and application-based directions yet to be explored in data distillation. In what follows, we highlight and discuss such directions for the community to further explore, based either on early evidence or our intuition.

**New data modalities.** Extending the discussion in Section 3, existing data distillation techniques have largely been restricted to cater to image datasets, primarily due to the amenable data-optimization in the continuous pixel-domain of images. Despite recent efforts in increasing the horizon of data distillation to other data modalities such as graphs (Jin et al., 2022a;b) and recommender systems (Sachdeva et al., 2022a); each data modality poses its unique challenges and calls for future work, *e.g.*, handling long sequences of time-series data in audio-classification (Hershey et al., 2017), video classification (Karpathy et al., 2014), self-driving (Sun et al., 2020); millions of categorical features in tabular data (Wang et al., 2021); sparse and noisy financial data (Xu & Cohen, 2018); *etc.*

**New predictive tasks.** Another limitation of existing data distillation techniques is that their underlying optimization is primarily designed for classification scenarios. However, a large number of predictive tasks fail to naïvely fit into the existing supervised data distillation framework, *e.g.*, image-generation (Ramesh et al., 2022; Rombach et al., 2022), language modeling (Brown et al., 2020; Devlin et al., 2019; Touvron et al., 2023), representation learning (Chen et al., 2020; Grill et al., 2020), *etc.* Further, the aforementioned tasks have gained immense popularity and have seen widespread practical use in the recent years, calling for future work in developing data distillation techniques for more predictive tasks.

**Better scaling.** Existing data distillation techniques validate their prowess *only* in the super low-data regime (typically $1 - 50$ data points per class) due to (i) computational difficulties in synthesizing large data summaries with existing techniques; and (ii) collapse to the random-sampling baseline when synthesizing large data summaries, as noted by Cui et al. (2022a). This calls for future work from both directions — developing efficient data distillation techniques that are scalable to web-scale datasets, and deeper investigations of the cause and potential fixes of the observed scaling artifacts of existing techniques.

**Improved optimization.** A unifying thread across data distillation techniques is an underlying bilevel optimization, which is provably NP-hard even in the linear inner-optimization case (Vicente et al., 1994). Notably, bilevel optimization has been successfully applied in a variety of other applications like meta-learning (Finn et al., 2017; Li et al., 2017), hyper-parameter optimization (Lorraine et al., 2020; Maclaurin et al.,

2015), neural architecture search (Liu et al., 2019), coreset construction (Borsos et al., 2020; Zhou et al., 2022a), *etc.* Despite its success, many theoretical underpinnings are yet to be explored, *e.g.*, the effect of commonly-used singleton solution assumption (Franceschi et al., 2018), the effect of over-parameterization on bilevel optimization (Vicol et al., 2022), connections to statistical influence functions (Bae et al., 2022), the bias-variance tradeoff (Vicol et al., 2021), *etc.* Clearly, an overall better understanding of bilevel optimization will directly enable the development of better data distillation techniques.

**Improved data-quality evaluation.** As briefly discussed in Section 2, data synthesized using data distillation is evaluated from performance, efficiency, and transferability standpoints. However, numerous high-stakes use-cases call for being able to train robust models from a variety of angles such as fairness (Mehrabi et al., 2021), adversarial robustness (Madry et al., 2018), *etc.* Hence, synthesizing data summaries able to support such robust model training is practical and an important direction for future work. Notably, while popular metrics exist for evaluating the robustness of learning algorithms from the aforementioned standpoints, developing such notions at the dataset-level is non-trivial, and with little existing literature (Ben-Eliezer & Yogev, 2020; Celis et al., 2018).

## Acknowledgments

We sincerely thank Zhiwei Deng, Bo Zhao, and George Cazenavette for their feedback on early drafts of this survey.

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

# A   Notation

### Dataset related

| | |
|---|---|
| $\mathcal{D} \triangleq \{(x_i \in \mathcal{X}, y_i \in \mathcal{Y})\}_{i=1}^{|\mathcal{D}|}$ | The target dataset to be distilled |
| $\mathcal{X}$ | Data domain |
| $\mathcal{Y}$ | Predictand domain |
| $\mathcal{C}$ | Set of unique classes in $\mathcal{Y}$ |
| $\mathcal{D}^c \triangleq \{(x_i, y_i) \mid y_i = c\}_{i=1}^{|\mathcal{D}|}$ | Portion of $\mathcal{D}$ with class $c$ |
| $\mathbf{X} \triangleq [x_i]_{i=1}^{|\mathcal{D}|}$ | Matrix of all features in $\mathcal{D}$ |
| $\mathbf{Y} \triangleq [y_i]_{i=1}^{|\mathcal{D}|}$ | Matrix of all predictands in $\mathcal{D}$ |
| $n$ | Size of data summary |
| $\mathcal{D}_{\mathsf{syn}} \triangleq \{(\tilde{x}_i, \tilde{y}_i)\}_{i=1}^{n}$ | Data summary |
| $\mathcal{D}_{\mathsf{syn}}^c \triangleq \{(\tilde{x}_i, \tilde{y}_i) \mid \tilde{y}_i = c\}_{i=1}^{n}$ | Portion of $\mathcal{D}_{\mathsf{syn}}$ with class $c$ |
| $\mathbf{X}_{\mathsf{syn}} \triangleq [\tilde{x}_i]_{i=1}^{n}$ | Matrix of all features in $\mathcal{D}_{\mathsf{syn}}$ |
| $\mathbf{Y}_{\mathsf{syn}} \triangleq [\tilde{y}_i]_{i=1}^{n}$ | Matrix of all predictands in $\mathcal{D}_{\mathsf{syn}}$ |

### Learning related

| | |
|---|---|
| $\Phi_\theta : \mathcal{X} \mapsto \mathcal{Y}$ | Learning algorithm parameterized by $\theta$ |
| $l : \mathcal{Y} \times \mathcal{Y} \mapsto \mathbb{R}$ | Twice-differentiable cost function |
| $\mathcal{L}_{\mathcal{D}}(\theta) \triangleq \mathbb{E}_{(x,y)\sim\mathcal{D}}[l(\Phi_\theta(x), y)]$ | Expected loss of $\Phi$ on $\mathcal{D}$ |
| $\mathcal{L}_{\mathcal{D}_{\mathsf{syn}}}(\theta) \triangleq \mathbb{E}_{(x,y)\sim\mathcal{D}_{\mathsf{syn}}}[l(\Phi_\theta(x), y)]$ | Expected loss of $\Phi$ on $\mathcal{D}_{\mathsf{syn}}$ |

### General

| | |
|---|---|
| $\dim(\mathcal{A})$ | Size of basis of $\mathcal{A}$ |
| $|\mathcal{A}|$ | Number of elements in $\mathcal{A}$ |
| $\sup$ | Supremum |
| $\arg\min_{\theta} f(\theta)$ | Optimum value of $\theta$ which minimizes $f(\theta)$ |
| $\mathbb{E}_x[f(x)] \triangleq \sum_x p(x) \cdot f(x)$ | Expected value of $f(x)$ when domain of $x$ is discrete |

