# OpenReview forum: "Data Distillation: A Survey"
_TMLR — Accepted by TMLR_

### Review · Reviewer_MxvN · 2023-04-24

**Summary Of Contributions:**

The contribution of this survey paper is to present a formal framework for data distillation and provide a detailed taxonomy of existing approaches. It also covers data distillation approaches for various data modalities, identifies current challenges, and proposes future research directions.

**Audience:**

Yes

**Broader Impact Concerns:**

This paper does not discuss any broader impact concerns.

**Claims And Evidence:**

Yes

**Requested Changes:**

- A taxonomy figure summarizing existing methods would be useful for readers to locate and compare different studies.
- The paper could indicate whether there is related work on data distillation.
- The results presented are limited to image modalities, and it would be beneficial if the paper discussed other modalities.
- The paper could benefit from a more detailed algorithm explaining the overall flow of the most basic model.
In the applications section, it would be helpful to define and quantify the results in each field. For example, for NAS, it would be useful to mention the scale of small and large NAS test-beds, whether they refer to the number of models in the search space, model size, or exploration time.

**Strengths And Weaknesses:**

**Strengths:**

- The paper's classification of different methods based on definitions, algorithms, and data modalities is clear and systematic, making it a valuable survey.
- The paper's categorization of applications, challenges, and future directions is also clear and insightful.

**Weaknesses:**

- The paper could benefit from a taxonomy figure summarizing existing methods, which would make it easier for readers to locate and compare different studies.
- It would be helpful if the paper indicated whether there is related work on surveys of data distillation.
- The results presented are limited to image modalities, and it would be beneficial if the paper discussed other modalities.
- The paper would benefit from a more detailed algorithm explaining the overall flow of the most basic model.
- In the applications section, it would be helpful to define and quantify the results in each field. For example, for NAS, it would be useful to mention the scale of small and large NAS test-beds, whether they refer to the number of models in the search space, model size, or exploration time.

---

> ### Author Response · Authors · 2023-06-13
> **Response to questions raised**
>
> We sincerely thank the reviewer for the time and effort in reviewing our paper and providing valuable feedback.
>
> We address each of the pointed weaknesses below:
>
> - Thank you for the suggestion — we will include a figure detailing the taxonomy of the methods discussed in the survey in our next revision.
> - As we note in our response to reviewer pzQW, our survey (submitted January 10, 2023) was the first ever survey on data distillation. In our humble opinion, it doesn’t ethically make sense to cite later work released on arXiv during the review process (~5 months).
> - We specifically discuss data distillation techniques for other data modalities in Section 3 and also point out challenges for expanding to newer data modalities and predictive tasks (Section 5). If you’re referring to an empirical comparison of data distillation techniques for non-image settings (viz. Table 1), we would like to point out that there exist only 1-2 data distillation techniques for such settings (recommender systems and graphs), making the point of comparing various data distillation techniques amongst each other quite trivial and non-informative.
> - Thank you for the suggestion — we will present an outline of the overall data distillation procedure as an algorithm in our next revision. It would complement nicely to the technicalities discussed in the paper.
> - Thank you for the suggestion again. While there are some irregularities in the NAS setups of various data distillation techniques prohibiting a head-to-head empirical comparison, we will elicit further details and subtle differences as suggested in our next revision.

---

### Review · Reviewer_hCkx · 2023-05-30

**Summary Of Contributions:**

This paper is a review on Data Distillation, which aims to reduce the size of datasets of various learning tasks and synthesize terse data summaries that can be used as alternatives for modeling training/inference. The paper begins by presenting the concept and rationale behind data distillation. Subsequently, it delves into a thorough examination of five primary categories of data distillation methods. Furthermore, the paper explores the modality and application of data distillation, providing valuable insights. Lastly, it outlines several prospective directions for future research in this field.

**Audience:**

Yes

**Broader Impact Concerns:**

I have no concerns on the broader impact.

**Claims And Evidence:**

Yes

**Requested Changes:**

Please attempt to address the concerns within the "Weakness" section.

**Strengths And Weaknesses:**

Strengths:
1. The topic discussed in the paper is highly significant, especially in the context of the prevalence of large-scale models today. With the escalating costs associated with model training, utilizing data distillation to expedite both training and inference processes is very important. This review has the potential to advance this objective.

2. The paper provides a concise and accurate summary of the existing five data distillation methods. It includes a comprehensive summary of primary references and offers method descriptions that are not only mathematically rigorous but also provide sufficient details.

3. The future directions outlined in Section 5 are valuable references for subsequent research in the field.

4. The paper is well-organized and written-well.


Weaknesses:
1. It seems that the paper only focused on labeled datasets. However, regarding tasks without annotations, such as unconditioned image synthesis, are there any works related to data distillation?

2. Currently, large-scale models and datasets are highly popular. Could the author summarize if there are any data distillation methods specifically designed for large datasets?

3. On Page 3, below Definition 3, the paper highlights three key criteria for evaluating data distillation methods: performance, efficiency, and transferability. I'm curious if robustness should be considered as a fourth criterion. For instance, when training with a distilled dataset, is the model more vonerable to adversarial attack methods?

4. An important role of data distillation is to boost training. Could the author provide a summary of the existing methods and their impact on training speed improvement?

---

> ### Author Response · Authors · 2023-06-13
> **Response to questions raised**
>
> We sincerely thank the reviewer for the time and effort in reviewing our paper and providing valuable feedback.
>
> We address each of the pointed weaknesses below:
>
> 1. You’re right — all existing data distillation techniques are designed for classification tasks, and developing data distillation techniques suitable for other predictive tasks (e.g., generation) is a promising future direction. Since there exists no work on data distillation for non-classification settings, we have already listed this direction (in addition to others) in the “New data modalities & settings” discussion in “Section 5: Challenges & Future Directions”.
>
> 2. While this is another promising direction for future research that is unexplored, existing techniques are limited to only the smaller academic datasets (e.g., CIFAR10, ImageNet subsets). We already hint toward this mismatch in the “Better scaling” discussion in “Section 5: Challenges & Future Directions”, but to make the connection more clear, we will reframe this in our next revision.
>
> 3. Thanks for the suggestion — it would indeed be an interesting idea to evaluate the synthesized data’s robustness. We promise to include a new discussion on “Robust data quality evaluation” in “Section 5: Challenges & Future Directions”.
>
> 4. While one of the major merits to data distillation is indeed to accelerate model training, existing work doesn’t explicitly mention the training speedup, simply because of a $1:1$ train-time speedup relative to the size of the training dataset. For example, if we distill CIFAR10’s train-set ($50k$ images) into a data summary of 10 images per class ($\equiv10\times10$ total images), we will observe a $\frac{50\cdot10^3}{100} = 500\times$ train-time speedup as the underlying original vs. distilled image size is the same. Since this seems like an important point to note, we will clarify this $1:1$ correlation in Section 2.

---

### Review · Reviewer_pzQW · 2023-06-13

**Summary Of Contributions:**

The paper surveyed methods, data modalities and applications of a newly emerging machine learning area called data distillation. The goal of data distillation is to synthesize small-scale data summaries from a large-scale dataset, which can act as drop-in replacement of the original dataset in training or inference scenarios. The author first proposed the concept of \epsilon-approximate data summary and defined data distillation as an optimization problem. Existing data distillation algorithms can be encapsulated in the framework as different ways of solving the optimization problem. The author grouped data distillation algorithms into five categories in the survey: 1) model matching, 2) gradient matching, 3) trajectory matching, 4) distribution matching, and 5) factorization. Afterwards, the author discussed the modalities that existing algorithms have been applied on, including image, text, graph and recommender systems. The author also introduced the applications that can benefit from the high-fidelity data summary, including differential privacy, neural architecture search, federated learning. In the end, the author briefly discussed the challenges in this area.

The paper is well-written, the unified framework is neat and the survey has included most of the recent publications.


**Audience:**

Yes

**Broader Impact Concerns:**

No concern

**Claims And Evidence:**

Yes

**Requested Changes:**

1. The author needs to clarify the relationship / difference of the paper with "A Survey on Dataset Distillation: Approaches, Applications and Future Directions".
2. The author should discuss the relationship between data distillation and data pruning
3. Most data distillation algorithms are still compared on classification datasets. The author should clarify such limitation and point out potential directions to expand data distillation to more tasks.


**Strengths And Weaknesses:**

Strengths
- The author proposed a unified framework for data distillation and surveyed recent algorithms. In addition, the data distillation methods are compared in Table 1.
- The author not only discussed the algorithms, but also listed the applications that can benefit from data distillation.

Weaknesses
- The author has not cited other surveys about data distillation, including "A Survey on Dataset Distillation: Approaches, Applications and Future Directions" which has been accepted by IJCAI 2023, and "A Comprehensive Survey of Dataset Distillation" (https://arxiv.org/pdf/2301.05603.pdf). In fact, the paper is quite similar to the IJCAI2023 survey paper For example, the IJCAI2023 survey paper also discussed model matching, gradient matching, trajectory matching, and distribution matching algorithms. One exception is that the previous survey paper has not discussed factorization-based data distillation methods. In addition, the IJCAI 2023 paper surveyed different data modalities and has included the audio modality, which was not included in the paper under review. The IJCAI 2023 paper also discussed different applications.
- The author compared data distillation with data compression in Section 2.5. I think the author may also compare data distillation with data pruning, especially discuss the relationship with recent paper like "Beyond neural scaling laws: beating power law scaling via data pruning".
- If we check the data modality section, we may notice that most problems are still classification problems. Applying data distillation to other tasks like language modeling or representation learning should also be a reasonable direction and the author may like to include it the survey.

---

> ### Author Response · Authors · 2023-06-13
> **Response to the questions raised**
>
> We sincerely thank the Reviewer for the time and effort in reviewing the paper and providing reasonable avenues to improve the survey.
>
> We strongly agree that the survey will benefit from (i) a more detailed comparison with data pruning studies (e.g., the Sorscher et.al paper referenced in the review) which we already cite albeit in a more abstract way; and (ii) a more explicit wording of how various data modalities discussed in the survey are also accompanied with different predictive tasks (vs. classification). We will make sure to address both of these suggestions in our next revision.
>
> However, we respectfully disagree with having to reason about other surveys on data distillation simply because all of the surveys were released **after** our submission (and our corresponding submission to arXiv). For example, even the referenced IJCAI’23 survey’s arXiv version was uploaded on 3rd of May whereas we submitted our paper to TMLR on 10th of January (~5 months from today) and was the first-ever survey on Data Distillation.
>
> We strongly believe the opposite case should be rather enforced where corresponding surveys cite this one, if such a way exists.

---

### Author Response · Authors · 2023-06-14
**Paper revision**

We firstly extend our heartfelt appreciation to the esteemed reviewers for generously dedicating their time and providing invaluable feedback. We have diligently taken into account all the suggestions put forth in the reviews and subsequently revised our manuscript. To facilitate convenience and transparency, we present a comprehensive changelist and a corresponding legend, clearly delineating the modifications made to address each reviewer's suggestions. We eagerly welcome any additional thoughts or suggestions to further refine and augment our survey.

*Legend for reviewer reference:*

**Reviewer pzQW**:  Changes #1, 2, 7

**Reviewer hCkx**:  Changes #3, 7, 8, 9

**Reviewer MxvN**:  Changes #4, 5, 6

*Changelist:*

1. **(Page 1, Section 1, Paragraph 1)** Enhanced the clarity of language to emphasize the impact of data quality on neural scaling laws.
2. **(Page 2, Section 1)** Introduced a new paragraph titled "Comparison with data pruning" to provide a comprehensive comparison between data distillation and data pruning/sampling/coreset construction techniques.
3. **(Page 3, Section 2, Paragraph 1)** Added a note highlighting data distillation’s linear training time speedup in relation to the data compression ratio.
4. **(Page 4, Figure 2)** Included a new figure that presents a taxonomy of data distillation techniques.
5. **(Page 5, Algorithm 1)** Incorporated an algorithm/pseudocode that describes the control flow of data distillation using the naÏve meta-matching framework.
6. **(Pages 12-13, Section 4)** Provided additional clarifications within the "Neural Architecture Search (NAS)" paragraph, specifically addressing the NAS test beds employed in existing studies.
7. **(Page 13, Section 5)** Restructured the "New data modalities & settings" paragraph into two distinct paragraphs: "New data modalities" and "New predictive tasks," with additional details about predictive tasks to which existing data distillation frameworks cannot be applied.
8. **(Page 13, Section 5)** Strengthened the wording in the "Better scaling" paragraph to emphasize the lack of scalable data distillation techniques.
9. **(Page 14, Section 5)** Introduced a new paragraph titled "Improved data quality evaluation" that explores potential future directions for robust evaluation of data summaries, including aspects such as fairness and adversarial robustness.

---

### Decision · Action_Editors · 2023-07-15

**Recommendation:** Accept as is

**Comment:**

This paper surveys methods, data modalities and applications of a newly emerging machine learning area called data distillation. The author first proposed the concept of epsilon-approximate data summary and defined data distillation as an optimization problem. The paper is well-written, the unified framework is neat and the survey has included most of the recent publications.

Based on three qualified reviews, this paper can be accepted and the authors are encouraged to merge the comments into their update versions.

**Audience:**

Yes

**Claims And Evidence:**

Yes